**Data Availability Statement:** The trial protocol and dataset supporting the conclusions of this article are available via the London School of Hygiene &

# COVID-19 self-testing using antigen rapid diagnostic tests: Feasibility evaluation among health-care workers and general population in Malawi

Madalo Mukoka [1,2] *, Euphemia Sibanda[3], Constancia Watadzaushe[3], Moses Kumwenda[1,2], Florence Abok[4], Elizabeth L. Corbett [5], Elena Ivanova[4], Augustine Talumba Choko[3]

1 Department of Pathology, Helse-Nord Tuberculosis Initiative, Kamuzu University of Health Sciences, Blantyre, Malawi, 2 Public Health and Policy Group, Malawi Liverpool Wellcome Trust Clinical Research Programme, Blantyre, Malawi, 3 The Centre for Sexual Health and HIV/AIDS Research Zimbabwe, Harare, Zimbabwe, 4 Foundation for Innovative New Diagnostics, Geneva, Switzerland, 5 Clinical Research Department, London School of Hygiene & Tropical Medicine, London, United Kingdom

* madalo.mukoka@gmail.com

## Abstract

### Background

COVID-19 testing is critical for identifying cases to prevent transmission. COVID-19 self-testing has the potential to increase diagnostic testing capacity and to expand access to hard-to-reach areas in low-and-middle-income countries. We investigated the feasibility and acceptability of COVID-19 self-sampling and self-testing using SARS-CoV-2 Antigen-Rapid Diagnostic Tests (Ag-RDTs).

### Methods

From July 2021 to February 2022, we conducted a mixed-methods cross-sectional study examining self-sampling and self-testing using Standard Q and Panbio COVID-19 Ag Rapid Test Device in Urban and rural Blantyre, Malawi. Health care workers and adults (18y+) in the general population were non-randomly sampled.

### Results

Overall, 1,330 participants were enrolled of whom 674 (56.0%) were female and 656 (54.0%) were male with 664 for self-sampling and 666 for self-testing. Mean age was 30.7y (standard deviation [SD] 9.6). Self-sampling usability threshold for Standard Q was 273/333 (82.0%: 95% CI 77.4% to 86.0%) and 261/331 (78.8%: 95% CI 74.1% to 83.1%) for Panbio. Self-testing threshold was 276/335 (82.4%: 95% CI 77.9% to 86.3%) and 300/332 (90.4%: 95% CI 86.7% to 93.3%) for Standard Q and Panbio, respectively. Agreement between self-sample results and professional test results was 325/325 (100%) and 322/322 (100%) for Standard Q and Panbio, respectively. For self-testing, agreement was 332/333 (99.7%: 95% CI 98.3 to 100%) for Standard Q and 330/330 (100%: 95% CI 99.8 to 100%) for Panbio.

Tropical Medicine Data Compass https://
datacompass.lshtm.ac.uk/.

**Funding:** This research was funded by UNITAID
(grant number KFW P09022-00) through the
Foundation For Innovative New Diagnostics. The
funder had no role in the study design, data
collection, analysis, decision to publish or
preparation of the manuscript.

**Competing interests:** The authors have declared
that no competing interests exist.

Odds of achieving self-sampling threshold increased if the participant was recruited from an urban site (odds ratio [OR] 2.15 95% CI 1.44 to 3.23, $P < .01$. Compared to participants with primary school education those with secondary and tertiary achieved higher self-testing threshold OR 1.88 (95% CI 1.17 to 3.01), $P = .01$ and 4.05 (95% CI 1.20 to 13.63), $P = .02$, respectively.

## Conclusions

One of the first studies to demonstrate high feasibility and acceptability of self-testing using SARS-CoV-2 Ag-RDTs among general and health-care worker populations in low- and middle-income countries potentially supporting large scale-up. Further research is warranted to provide optimal delivery strategies of self-testing.

## Introduction

Only around 0.2% of people in Africa had tested for severe acute respiratory syndrome coronavirus 2 (SARS-CoV-2) the infection that causes COVID-19 in August 2020 [1] By contrast, 19.5% of Americans had tested by the same time [1] since COVID-19 emergence in December 2019 [2, 3]. These contrasting trends have continued to exist with widening unequal access to testing, treatment and vaccination between high income countries and low- and middle-income countries (LMICs) despite four or more global epidemic waves [4]. However, the access gap has become narrower as the pandemic has grown older [5]. Testing remains the most critical step for identification and isolation of COVID-19 cases to prevent transmission [6]. In many resource-limited settings, demand for tests often exceeds supply [7]. SARS-CoV-2 rapid antigen tests (Ag-RDTs) are recommended to complement nucleic acid amplification tests (NAAT) for diagnosis [8], which in resource-limited settings are often hard to implement because they require specialised skills and limited centralized laboratory capacity, associated with long turnaround times, and high costs to both the health system and patients [9, 10].

COVID-19 self-testing was strongly recommended by the World Health Organization (WHO) in March 2022 as an additional strategy to complement professionally administered testing services [11]. Self-sampling and self-testing is a process by which a person collects his or her own specimen using a simple device, performs a diagnostic test and interprets the results usually in a setting, and time of their choice [12]. Self-testing is not a new paradigm with pregnancy self-testing and HIV self-testing being successful examples [13–16]. In general, COVID-19 self-testing has the potential to increase diagnostic capacity for COVID-19 and reduce access barriers as well as prevailing inequalities due to ease of distribution and being extremely convenient [17]. However, COVID-19 self-testing has so far been widely implemented and made available in high income countries with reported high feasibility and acceptability [17–21]. According to our knowledge, there were no reports on COVID-19 self-testing feasibility in LMICs prior to this study. As with HIV self-testing, lack of linkage for next steps with COVID-19 is a potential concern due to stigma, loss of economic opportunities due to isolation implications, and fear of complications including death.

Being able to self-test rests on the assumption that individuals would be able to take their own sample (self-sampling) [22]. However, in settings with low exposure to technology and the ability to correctly follow instructions such assumptions may be faulty [23]. Thus, early work including optimization of instructions for use through iterative cognitive interviews is

essential to ensure correct use of self-tests [23]. Here we investigated the feasibility and acceptability of COVID-19 self-sampling and self-testing using SARS-CoV-2 Ag-rapid diagnostics tests (RDTs) in Malawi.

## Materials and methods

### Study design

A mixed-methods cross-sectional study examining self-sampling and self-testing for COVID-19 using STANDARD Q COVID-19 Ag Test (SD Biosensor) and Panbio COVID-19 Ag Rapid Test Device (Abbott Rapid Diagnostics). We conducted the study under five components. They were conducted serially as follows: cognitive interviews to refine instructions for use (IFUs) for self-sampling, observational cross-sectional study of self-sampling, cognitive interviews to refine instructions for use for self-testing, observational cross-sectional study of self-testing, and in-depth interviews (IDIs) to understand participant views on self-sampling and self-testing.

### Setting

Recruitment was conducted between July 2021 to February 2022 from Queen Elizabeth Central Hospital (QECH) from urban Blantyre, Malawi (Fig 1) and from Lirangwe Primary Health Centre from rural Blantyre (Fig 2). Sites were chosen to ensure inclusion of both rural and

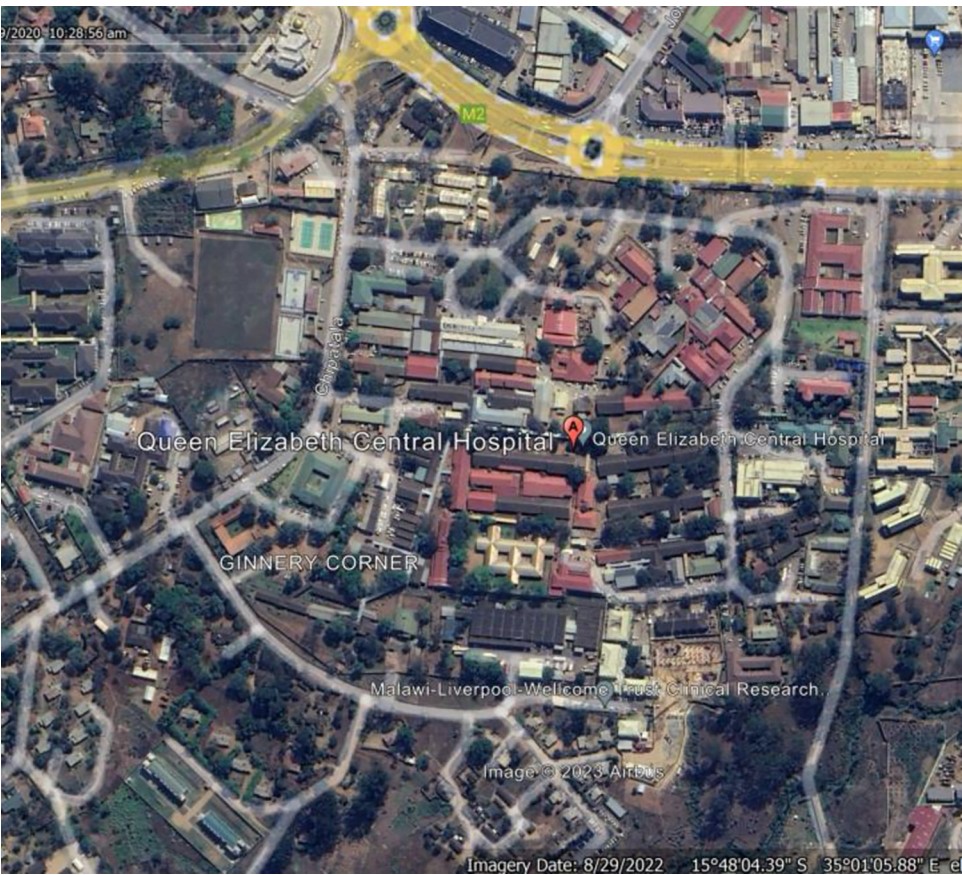

**Fig 1. Map of Blantyre with Queen Elizabeth Central Hospital marked.**

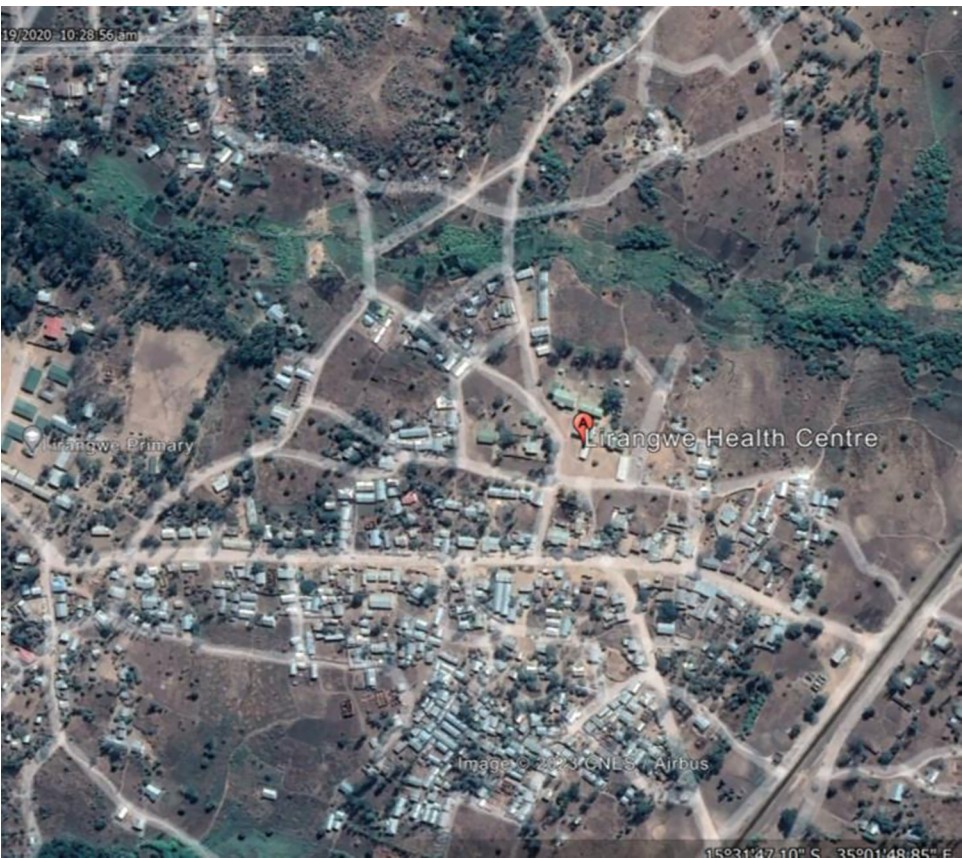

**Fig 2. Map of Blantyre with Lirangwe health centre marked.**

urban participants whose literacy levels, technology exposure and perception to testing may differ. At the time of recruitment, COVID-19 testing in Blantyre was concentrated at QECH whereas Lirangwe health centre was only collecting samples to be tested at another facility.

## Participants

We recruited health care workers and members of the general public from the recruitment sites. To be eligible, participants needed to be 18 years or older, feeling well enough to comfortably conduct study activities, not having recent history of excessive nose bleeds, and having given consent. All health workers from the two health facilities were offered the choice to participate in the study with exclusion only done if ineligible. An additional eligibility criteria which was later relaxed due to scarcity of participants with waning wave concerned individuals being on the list to be tested for COVID-19 by the national systems. General public participants were non randomly sampled from outpatient departments. An anterior nasal swab for COVID-19 was done for both self-sampling and self-testing following a short in-person demonstration by a member of staff.

Participants were observed in-person during self-sampling and self-testing, and checklists (Tables A-D in S1 text) were completed to document whether each task was done correctly. A trained researcher then tested the collected sample using a SARS-CoV-2 Ag RDT during the self-sampling component of the study. Participants tested their own collected sample during the self-testing component of the study. The trained researcher collected and tested an anterior

nasal confirmatory sample using an Ag RDT during both the self-sampling and self-testing components.

## Variables

For the cognitive interviews, the main output was to have refined IFUs in the local language (Chichewa) and in English. The first primary outcome was the percentage of participants who attained a usability threshold for self-sampling, defined as correct execution of all critical instructions during the self-sampling process for each kit. Correct self-sampling was referred to as self-sampling accuracy. The second primary outcome was the percentage of participants who attained a usability threshold for self-testing, defined as correct execution of all critical instructions during the self-testing process. Correct self-testing was referred to as self-testing accuracy. User views regarding self-sampling and self-testing were the main outcomes from the IDIs. Potential confounders for accuracy were age, sex, literacy and prior exposure to COVID-19 testing.

## Data sources/ measurement

Qualitative data from cognitive interviews and IDIs were tape recorded before being translated and transcribed. Pre- and post-test questionnaires were administered in-person using Open Data Kit (ODK) loaded on tablets. A checklist (Tables A-D in S1 text) was completed by a member of staff to document whether each instruction was done correctly as a measure of accuracy. Results obtained by a trained researcher from testing the collected self-sample and a sample collected by the researcher were recorded on the checklist. For self-testing accuracy, participant's self-test self-read results were compared to RDT sampling and testing conducted by the researcher. Participants' reading of pre-made cassettes of negative, positive and invalid results was also recorded.

## Bias

The main source of bias is in the assessment by the research staff using a checklist of the performance of the participant on the IFU. A staff member who was more punitive may have harshly rated performance as incorrect while a more forgiving one may have rated performance differently. However, the fact that more than seven staff members were involved in the rating may have minimized such bias.

## Study size

We aimed to recruit and purposively sample 120 participants for cognitive interviews for self-sampling and self-testing for both test kits. For self-sampling and self-testing, we conservatively assumed that 70% to 80% of participants will be able to correctly follow instructions and self-sample or self-test for COVID-19. For the sample proportion to be estimated to within +/-0.05 (5%) using the 95% confidence level, a sample of 323 participants were required. Thus, a total of 1,320 participants were needed: 330 per test kit for self-sampling and self-testing. A purposive sample of 120 participants was needed for the IDIs: 60 self-sampling and 60 self-testing participants.

## Quantitative variables

A binary variable was generated for the first and secondary primary outcomes of achieving the threshold (accuracy) for either self-sampling or self-testing. This was coded as 1 for participants with a maximum score on the critical steps based on the checklist and 0 otherwise. Test result variables were coded as 1 for positive and 0 for negative.

## Statistical methods

Analysis used R [24]with 0.05 as an indicator of statistical significance. Frequencies were computed for categorical variables while mean and standard deviation (SD) or median and (inter quartile range) were computed for continuous variables that were normally distributed or skewed, respectively. We computed the proportion achieving accuracy along with Binomial Exact confidence intervals (CIs) for self-sampling and self-testing for each test kit. Similarly, we computed the proportion of self-test results that agreed with staff conducted RDT test results along with Binomial Exact confidence intervals (CIs) for each test kit. Logistic regression was used to examine factors associated with accuracy.

## Ethics statement

The study was conducted according to the guidelines of the Declaration of Helsinki, and approved by the Malawi College of Medicine Research Ethics Committee of Kamuzu University of Health Sciences (Reg No: P.03/21/3277) and the World Health Organization Research Ethics Review Committee (Protocol ID: CERC.0104). Informed consent was obtained from all participants involved in the study.

# Results

## Participants and descriptive data

A total of 1,510 participants were recruited across the five components of the study. A total of 120 participants were recruited for self-sampling and self-testing cognitive interviews (self-sampling (n = 76) and self-testing (n = 44)). Of 723 screened for eligibility 664 (91.8%) were recruited for self-sampling with mean age of 31.4y (standard deviation [SD]: 9.8) and 357/664 (53.8%) were male (Table 1). For self-testing, 666 (95.4%) were recruited of 698 screened for eligibility; mean age was 30.6y (standard deviation [SD]: 9.6) with 293/666 (44.0%) being male (Table 2). The main exclusion was being under 18 years. Sixty participants were recruited for IDIs.

## Outcome data

The cognitive interviews showed that participants in both rural and urban communities were able to follow the IFUs with no major suggestions for changes. Notable changes to IFUs included: making introductory text stand out to catch attention, enhancing clarity of IFUs such as by expanding text, adding labels on images, selecting words or phrases that could be well understood locally. Insertion of test swab to correct depth (1.5cm or 2cm) was illustrated by reference to inserting up to thumbnail depth.

## Main results

Self-sampling accuracy was 273/333 (82.0%: 95% CI: 77.4 to 86.0) for Standard Q and 261/331 (78.8: 95% CI: 74.1% to 83.1%) for Panbio (Table 3). The percentage agreement between the test results from the participant and the study staff was 100% for both kits in Malawi (Table 3).

Self-testing accuracy was 276/335 (82.4%: 95% CI: 77.9 to 86.3) for Standard Q and 300/332 (90.4%: 95% CI: 86.7 to 93.3) for Panbio (Table 3). The percentage agreement between the test results from the participant and the study staff was 99.7% (95% CI: 98.3–100%) for Standard Q with only one false negative self-test self-read result.

Up to 95% of the critical steps were performed correctly on either test kit for both self-sampling and self-testing (Table 4).

The odds of self-sampling accuracy increased 2-fold for participants from QECH compared to participants from Lirangwe primary health centre odds ratio (OR) 2.15 (95% CI 1.44 to

**Table 1. Baseline characteristics: Self-sampling.**

| Variable | Characteristic | Overall | Standard Q | Panbio | p-value[a] |
|---|---|---|---|---|---|
| Number of participants | n | 664 | 331 | 333 | |
| Sex | Male | 357 (53.8) | 176 (53.2) | 181 (54.4) | 0.820 |
| | Female | 307 (46.2) | 155 (46.8) | 152 (45.6) | |
| Age (years) | mean (SD) | 31.4 (9.8) | 31.8 (10.3) | 31.0 (9.3) | 0.313 |
| Ever tested for COVID-19? | No | 568 (87.0) | 303 (91.8) | 265 (82.0) | <0.001 |
| | Yes | 85 (13.0) | 27 (8.2) | 58 (18.0) | |
| Marital status | Divorced | 37 (5.7) | 20 (6.1) | 17 (5.3) | 0.512 |
| | Separated | 33 (5.1) | 15 (4.5) | 18 (5.6) | |
| | Widowed | 14 (2.1) | 8 (2.4) | 6 (1.9) | |
| | Never married | 180 (27.6) | 82 (24.8) | 98 (30.3) | |
| | Married | 389 (59.6) | 205 (62.1) | 184 (57.0) | |
| Money earned per month (MWK) | mean (SD) | 68191 (96060) | 57471 (90035) | 79111 (100803) | 0.004 |
| Able to read a newspaper? | No | 42 (6.4) | 29 (8.8) | 13 (4.0) | 0.020 |
| | Yes | 610 (93.6) | 300 (91.2) | 310 (96.0) | |
| Highest level of formal schooling | Never been to school | 25 (3.9) | 18 (5.5) | 7 (2.2) | <0.001 |
| | Primary | 165 (25.4) | 101 (31.0) | 64 (19.8) | |
| | Secondary no MSCE | 216 (33.3) | 118 (36.2) | 98 (30.3) | |
| | Secondary with MSCE | 131 (20.2) | 54 (16.6) | 77 (23.8) | |
| | Tertiary | 112 (17.3) | 35 (10.7) | 77 (23.8) | |
| Number of people in household | mean (SD) | 4.2 (1.8) | 4.2 (1.8) | 4.2 (1.8) | 0.903 |
| Number of rooms in household | mean (SD) | 2.5 (1.0) | 2.4 (1.0) | 2.6 (1.1) | 0.032 |
| Number of households per dwelling | mean (SD) | 1.5 (1.3) | 1.3 (0.8) | 1.7 (1.7) | <0.001 |
| Enough food / essentials for 14 days? | No | 412 (63.1) | 215 (65.2) | 197 (61.0) | 0.307 |
| | Yes | 241 (36.9) | 115 (34.8) | 126 (39.0) | |
| Recruitment site | QECH | 331 (49.8) | 165 (49.8) | 166 (49.8) | 1.000 |
| | Lirangwe | 333 (50.2) | 166 (50.2) | 167 (50.2) | |

[a]Chisquare test for categorical variables; t-test for continuous variables

SD: standard deviation; QECH: Queen Elizabeth Central Hospital

3.23, $P < 0.001$ (Table 5). There appeared to be a linear trend towards increased odds of attaining self-testing accuracy with increasing levels of education, $P$ for trend 0.01.

## Other analyses

All in-depth interview participants reported that self-testing was highly acceptable because it was convenient, empowering and private.

Most participants had no problems interpreting contrived panel results with 99% correctly interpreting positive and negative results correctly although 96% correctly interpreted invalid results on either test kit (S1 Table). Up to 90.7% Standard Q and 96.1% Panbio participants found instructions "not at all hard" when asked on exit interviews (S2 Table).

## Discussion

### Key results

This is one of first studies, to the best of our knowledge, to be conducted on COVID-19 self-testing in low- and middle-income countries and generally indicates that participants in both rural and urban communities in Malawi can self-test correctly for COVID-19. The results of

**Table 2. Baseline characteristics: Self-testing.**

| Variable | Characteristic | Overall | Standard Q | Panbio | p-value[a] |
|---|---|---|---|---|---|
| **Number of participants** | **n** | **664** | **336** | **328** | |
| **Sex** | **Male** | **292 (44.0)** | **138 (41.1)** | **154 (47.0)** | **0.148** |
| | Female | 372 (56.0) | 198 (58.9) | 174 (53.0) | |
| Age (years) | mean (SD) | 30.7 (9.6) | 30.8 (9.8) | 30.52 (9.3) | 0.724 |
| Ever tested for COVID-19? | No | 603 (91.5) | 303 (91.0) | 300 (92.0) | 0.659 |
| | Yes | 56 (8.5) | 30 (9.0) | 26 (8.0) | |
| Marital status | Divorced | 37 (5.6) | 15 (4.5) | 22 (6.8) | 0.208 |
| | Separated | 47 (7.2) | 24 (7.2) | 23 (7.1) | |
| | Widowed | 25 (3.8) | 13 (3.9) | 12 (3.7) | |
| | Never married | 195 (29.7) | 88 (26.5) | 107 (32.9) | |
| | Married | 353 (53.7) | 192 (57.8) | 161 (49.5) | |
| Money earned per month (MWK) | mean (SD) | 67802 (123668) | 70359 (153192) | 65190 (83531) | 0.591 |
| Able to read a newspaper? | No | 59 (8.9) | 29 (8.7) | 30 (9.2) | 0.922 |
| | Yes | 601 (91.1) | 305 (91.3) | 296 (90.8) | |
| Highest level of formal schooling | Never been to school | 33 (5.0) | 19 (5.7) | 14 (4.3) | 0.209 |
| | Primary | 208 (31.5) | 109 (32.6) | 99 (30.4) | |
| | Secondary no MSCE | 220 (33.3) | 117 (35.0) | 103 (31.6) | |
| | Secondary with MSCE | 143 (21.7) | 68 (20.4) | 75 (23.0) | |
| | Tertiary | 56 (8.5) | 21 (6.3) | 35 (10.7) | |
| Number of people in household | mean (SD) | 4.2 (1.6) | 4.2 (1.7) | 4.3 (1.6) | 0.548 |
| Number of rooms in household | mean (SD) | 2.5 (1.2) | 2.5 (1.0) | 2.5 (1.4) | 0.559 |
| Number of households per dwelling | mean (SD) | 1.5 (1.3) | 1.5 (1.6) | 1.5 (0.9) | 0.516 |
| Enough food / essentials for 14 days? | No | 351 (53.3) | 188 (56.5) | 163 (50.0) | 0.113 |
| | Yes | 308 (46.7) | 145 (43.5) | 163 (50.0) | |
| Recruitment site | QECH | 334 (50.8) | 169 (51.1) | 165 (50.6) | 0.971 |
| | Lirangwe | 323 (49.2) | 162 (48.9) | 161 (49.4) | |

this study show that 82% and 90% of participants were able to self-test for COVID-19 with no supervision following a brief demonstration using Standard Q and Panbio test kits, respectively. Of further note, all self-test results agreed 100% with professionally conducted RDTs for Panbio kit whereas agreement was 99.7% for Standard Q. Similarly, 82% of participants were able to correctly self-sample for COVID-19 using Standard Q compared to 79% using Panbio. COVID-19 self-testing was rated as highly acceptable during in-depth interviews.

**Table 3. Self-sampling and self-testing accuracy.**

| | Standard Q | | | | | Panbio | | | | |
|---|---|---|---|---|---|---|---|---|---|---|
| | N | n | % | 95% CI | | N | n | % | 95% CI | |
| Met self-sampling threshold[a] | 333 | 273 | 82.0 | 77.4 | 86.0 | 331 | 261 | 78.8 | 74 | 83.1 |
| Met self-testing threshold | 335 | 276 | 82.4 | 77.9 | 86.3 | 332 | 300 | 90.4 | 86.7 | 93.3 |
| Agreement with professional test | | | | | | | | | | |
| Self-sampling | 322 | 322 | 100 | 99 | 100 | 325 | 325 | 100 | 100 | 100 |
| Self-testing | 333 | 332 | 99.7 | 98 | 100 | 330 | 330 | 100 | 100 | 100 |

[a]Threshold: participant performing all critical steps correctly

CI: confidence interval

**Table 4. User errors for Standard Q and Panbio kits.**

| | Standard Q (N = 331) | | Panbio (N = 331) | |
|---|---|---|---|---|
| | **Yes** | **No** | **Yes** | **No** |
| Did the participant place the tube on the kit box tray holder or flat surface correctly? | 327 (97.6) | 8 (2.4) | 329 (99.4) | 2 (0.6) |
| Did participant insert the swab into the left nostril to the correct depth (about 1.5cm or 2cm)? | 327 (97.6) | 8 (2.4) | 332 (100) | 0 (0.0) |
| Did the participant rotate the swab 5 or 10 times in the left nostril? | 325 (97.0) | 10 (3.0) | 327 (98.5) | 5 (1.5) |
| Did participant insert the swab into the right nostril to the correct depth (about 1.5cm or 2cm)? | 327 (98.5) | 5 (1.5) | 329 (99.4) | 2 (0.6) |
| Did the participant rotate the swab 5 or 10 times in the right nostril? | 324 (97.0) | 10 (3.0) | 330 (99.4) | 2 (0.6) |
| Did the participant insert the swab into the solution tube correctly? | 331 (98.8) | 4 (1.2) | 328 (99.4) | 2 (0.6) |
| Did the participant swirl in the fluid 5 or 10 times while pushing against the wall of the tube? | 325 (97.3) | 9 (2.7) | 323 (97.6) | 8 (2.4) |
| Did the participant remove the swab slowly while squeezing the sides of the tube to extract the liquid from the swab? | 310 (92.5) | 25 (7.5) | 316 (95.5) | 15 (4.5) |
| Did the participant press the nozzle cap tightly the tube? | 326 (97.9) | 7 (2.1) | 330 (99.7) | 1 (0.3) |
| Did the participant squeeze 4 or 5 drops of liquid from the tube into the well on the test device? | 319 (95.5) | 15 (4.5) | 330 (99.7) | 1 (0.3) |
| Did the participant read the test result in 15 minutes? | 333 (99.7) | 1 (0.3) | 325 (97.9) | 7 (2.1) |
| Did the participant interpret the test result correctly? | 328 (98.5)s | 2 (1.5) | 329 (99.7) | 1 (0.3) |

Standard Q: swab 10 times, depth 2cm, 4 drops

Current strategies for COVID-19 testing in high income countries are largely dependent on Ag-RDT self-sampling and self-testing [25–27] with over-the-counter self-test kits available for purchase in a wide range of countries [21, 28]. The limited data available in resource-poor settings suggest that, as with HIV self-testing, diagnostic accuracy is not as great with untrained lay users as with trained professionals, mainly affecting sensitivity [20, 25, 29]. Our results on the other hand show that self-testing accuracy improved markedly with a short demonstration supporting previous findings observed with HIV self-testing [14, 30]. However, there is still a place for well-translated and culturally relevant IFUs to support the large-scale implementation of self-testing. Our study investigated self-testing with two kits that were already approved for use in Malawi. However, there are numerous Ag-RDT tests packaged for COVID-19 self-testing that have met performance standards and been approved by Regulatory Authorities such as the FDA, that may yield similar promising results [31, 32].

## Limitations

There are notable limitations with our study. Firstly, there was a small number of positive self-test results. Although this does not affect the reading of correct results and indeed completing critical steps correctly as assessed here it may be important as it is likely to affect sensitivity [33]. Reassuringly, up to 99% of participants correctly interpreted contrived positive results on either kit. Secondly, there was potential for assessment bias resulting from subjective judgement on the checklist used by research staff for assessing performance of the participant on

**Table 5. Factors associated with self-sampling and self-testing accuracy.**

| Variable | Characteristic | Self-sampling (N = 641) | | | | Self-testing (N = 637) | | | |
|---|---|---|---|---|---|---|---|---|---|
| | | Unadjusted | | | | Unadjusted | | | |
| | | OR | 95% CI | | p-value | OR | 95% CI | | p-value |
| Age | Yearly increase | 0.99 | 0.97 | 1.01 | 0.247 | 0.99 | 0.97 | 1.01 | 0.299 |
| Sex | Female | 1.00 | | | | 1.00 | | | |
| | Male | 1.00 | 0.68 | 1.48 | 0.991 | 0.84 | 0.54 | 1.33 | 0.459 |
| Site | Lirangwe | 1.00 | | | | 1.00 | | | |
| | QECH | 2.15 | 1.44 | 3.23 | <0.001 | 1.47 | 0.93 | 2.32 | 0.097 |
| Literacy | No | 1.00 | | | | 1.00 | | | |
| | Yes | 0.57 | 0.22 | 1.49 | 0.251 | 1.21 | 0.57 | 2.55 | 0.625 |
| Highest level of education attained? | Primary school | 1.00 | | | | 1.00 | | | |
| | Never been school | 2.75 | 0.61 | 12.3 | 0.186 | 1.67 | 0.55 | 5.02 | 0.362 |
| | Secondary | 0.91 | 0.58 | 1.44 | 0.686 | 1.88 | 1.17 | 3.01 | 0.009 |
| | Tertiary | 2.09 | 1.03 | 4.25 | 0.041 | 4.05 | 1.20 | 13.63 | 0.024 |
| Marital Status | Divorced/separated/widowed | 1.00 | | | | 1.00 | | | |
| | Never married | 2.10 | 1.06 | 4.14 | 0.033 | 2.41 | 1.26 | 4.61 | 0.008 |
| | Married | 1.07 | 0.61 | 1.89 | 0.820 | 1.89 | 1.09 | 3.28 | 0.023 |
| Ever tested for COVID-19? | No | 1.00 | | | | 1.00 | | | |
| | Yes | 1.04 | 0.58 | 1.86 | 0.905 | 1.25 | 0.52 | 3.02 | 0.619 |

OR: odds ratio; CI: confidence interval; QECH: Queen Elizabeth Central Hospital

each instruction. The impact of this bias could be bi-directional depending on whether the staff was harsh–leading to poor rating, or more lenient resulting in more participants being passed as correctly following instructions.

## Generalisability

This study demonstrates high acceptability and feasibility of COVID-19 self-testing [11]. The findings are very similar to results reported in other self-testing areas including HIV [34] and hepatitis C virus (HCV) [35]. Thus, we posit that the findings are generalizable to many resource settings and populations including those with limited literacy. However, some support may be useful for specific settings and users–such as older age groups and those with lower literacy. Lessons learned from introduction and scale-up of other self-testing approaches such as HIV and HCV may be appliable here to accelerate adaptation plans and efforts in LMIC.

## Conclusions

This is one the first studies to demonstrate high usability and acceptability of self-testing using SARS-CoV-2 Ag-RDTs among both general and health-care worker populations in LMICs. While most users collected their own samples and self-tested with ease, participants noted demonstrations were helpful and could be important in some settings and populations, such as older age groups and those with low literacy levels. COVID-19 self-testing is an important strategy for further consideration as it may be a promising tool for increasing access to and uptake of COVID-19 testing services as well as strategies to reduce transmission and linkage to further care, treatment and support services. Further research is warranted to provide optimal delivery strategies to reach priority populations in LMICs.

## Supporting information

**S1 Text. Checklists for self-sampling and self-testing-Standard Q and Panbio.**
(DOCX)

**S1 Table. Interpreting contrived panel results.**
(DOCX)

**S2 Table. User views on self-sampling and self-testing.**
(DOCX)

## Acknowledgments

The authors would like to thank all study participants, participating health facilities, and project team members for their contributions to the successful completion of this project.

## Author Contributions

**Conceptualization:** Madalo Mukoka, Euphemia Sibanda, Constancia Watadzaushe, Moses Kumwenda, Florence Abok, Elizabeth L. Corbett, Elena Ivanova, Augustine Talumba Choko.

**Data curation:** Madalo Mukoka, Augustine Talumba Choko.

**Formal analysis:** Madalo Mukoka, Euphemia Sibanda, Moses Kumwenda, Augustine Talumba Choko.

**Funding acquisition:** Euphemia Sibanda, Florence Abok, Elizabeth L. Corbett, Elena Ivanova, Augustine Talumba Choko.

**Investigation:** Madalo Mukoka, Euphemia Sibanda, Constancia Watadzaushe, Moses Kumwenda, Florence Abok, Augustine Talumba Choko.

**Methodology:** Madalo Mukoka, Euphemia Sibanda, Constancia Watadzaushe, Moses Kumwenda, Florence Abok, Augustine Talumba Choko.

**Project administration:** Madalo Mukoka, Euphemia Sibanda, Constancia Watadzaushe, Moses Kumwenda, Florence Abok, Elizabeth L. Corbett, Elena Ivanova, Augustine Talumba Choko.

**Resources:** Madalo Mukoka, Euphemia Sibanda, Elena Ivanova, Augustine Talumba Choko.

**Software:** Augustine Talumba Choko.

**Supervision:** Euphemia Sibanda, Moses Kumwenda, Florence Abok, Elizabeth L. Corbett, Elena Ivanova, Augustine Talumba Choko.

**Validation:** Madalo Mukoka, Euphemia Sibanda, Elizabeth L. Corbett, Elena Ivanova, Augustine Talumba Choko.

**Visualization:** Euphemia Sibanda, Elizabeth L. Corbett, Elena Ivanova, Augustine Talumba Choko.

**Writing – original draft:** Madalo Mukoka, Augustine Talumba Choko.

**Writing – review & editing:** Madalo Mukoka, Euphemia Sibanda, Constancia Watadzaushe, Moses Kumwenda, Florence Abok, Elizabeth L. Corbett, Elena Ivanova.

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
