## [Decision Letter · Decision Letter 0]

27 Apr 2023

PONE-D-22-25536COVID-19 self-testing using antigen rapid diagnostic tests: feasibility evaluation among health-care workers and general population in MalawiPLOS ONE

Dear Dr. Mukoka,

Thank you for submitting your manuscript to PLOS ONE. I sincerely apologise for the unusually delayed review timeframe. Your manuscript has been assessed by two reviewers, whose comments are appended below (one of the reports is also attached as a PDF). The reviewers comment positively on the research question being investigated, but raise minor concerns regarding aspects of the methodology and reporting. After careful consideration, we feel that it has merit but does not fully meet PLOS ONE’s publication criteria as it currently stands. Therefore, we invite you to submit a revised version of the manuscript that addresses the points raised during the review process.

We look forward to receiving your revised manuscript.

Kind regards,

Emily Chenette

Editor in Chief

PLOS ONE

Journal Requirements:

2. Please ensure that you have specified (1) whether consent was informed and (2) what type you obtained (for instance, written or verbal, and if verbal, how it was documented and witnessed). If your study included minors, state whether you obtained consent from parents or guardians. If the need for consent was waived by the ethics committee, please include this information.

Reviewers' comments:

Reviewer's Responses to Questions

**Comments to the Author**

1. Is the manuscript technically sound, and do the data support the conclusions?

Reviewer #1: Yes

Reviewer #2: Yes

2. Has the statistical analysis been performed appropriately and rigorously? 

Reviewer #1: I Don't Know

Reviewer #2: Yes

3. Have the authors made all data underlying the findings in their manuscript fully available?

Reviewer #1: Yes

Reviewer #2: Yes

4. Is the manuscript presented in an intelligible fashion and written in standard English?

Reviewer #1: No

Reviewer #2: Yes

5. Review Comments to the Author

Reviewer #1: This is a well written study with explanatory study design. It will need language review to polish its novelty in an English journal.

I advised that "Keywords section be added to the abstract"

Other sectional concerns are as in the returned reviewed article.

Some minor reviews should bring the study to level suitable of publication in a standard medical journal.

Reviewer #2: Title: COVID-19 self-testing using antigen rapid diagnostic tests: feasibility evaluation among health-care workers and general population in Malawi

Well-done to the authors for such a robust work on Covid-19 self-testing and self sampling

Abstract

In the background and throughout the document- The use of COVID-19 and SARS-CoV-2 interchangeably is confusing. The context in which they are used can be more defined. I will suggest using COVID-19 for self-testing and sampling instances and SARS-CoV-2 for the rapid diagnostic test. For example, COVID-19 self-testing and COVID-19 sampling and then SARS-CoV-2 Antigen-Rapid Diagnostic Tests (Ag-RDTs).

Line 31- Begin methods with From July…….

Line 33- In the methods are the Standard Q and Panbio tests registered trademarks? If so indicate with the symbol®

Line 34- randomly or non-random sampling?

Line 48-49-The conclusion can include summary points for usability and acceptability of the tests. Also, implications for practice and policies?

Introduction

Line 61- are the SARS-CoV-2 rapid antigen tests (Ag-RDTs) recommended? By who? Or do you mean they are specifically useful to complement….

Line 64- would be good to provide refs for these testing setbacks in resource limited setting

Line 65- provide a definition of self-testing and self-sampling in the context in which it is being used in this paper

The introduction could be further strengthened re- what is the acceptability and feasibility of COVID-19 self-testing reported in HICs? Have there been any reported use of COVID-19 self-tests in other low resource settings prior? What is the general argument with regards to feasibility and acceptability of COVID-19 self-tests especially in LMICs?

Methods

Line 86- were the five components or phases conducted one after the other or in parallel or was there any specific order?

Line 104- how were the participants systematically sampled from out-patients? In-person during attendance? Via registers?

Line 126- Was participant consent obtained for audio recording?

Line 155- How was the qualitative data from interviews analysed and presented?

Line 134- Was there any bias introduced from researcher interpretation of interview data? Or influence in the results? Any steps taken to address reflexivity in the qualitative aspects?

Results

Line 182- Best to start with total number of participants included in the study and then present the breakdown of participants per component. Starting with a total of 120 participants and then going up to 664 recruited from 732 screened is confusing

Line 183 how m any were recruited from each site and how many were selected for interviews?

Line 189- How many participants were in the cognitive interviews?

Line 228- Nearly is how many out of how many total interviewed?

Discussion

Line 237- one of the first studies to the best of your knowledge?

Line 242- is it Panbio or PanBio? Review and maintain consistency throughout the manuscript.

Line 258- I don’t really see a small number of positive results as a limitation but a strength how was this managed? Linkage to care? If shows the tests were administered correctly

Line 264- were any steps taken to reduce the impact on extreme assessors?

Line 266- Are there others strengths of this work? Originality?

Line 274- What are the key recommendations? Any implications for practice policies and future research?

Line 276- missing “of” in 2 places

Line 277- LMICs if defined earlier no need to spell out in full

6. PLOS authors have the option to publish the peer review history of their article (what does this mean?). If published, this will include your full peer review and any attached files.

Reviewer #1: **Yes: **Adeloye A Adeniji

Reviewer #2: No

---

## [Author Response · Author response to Decision Letter 0]

8 Jun 2023

This is a well written study with explanatory study design. It will need language review to polish its novelty in an English journal.

I advised that "Keywords section be added to the abstract"

Other sectional concerns are as in the returned reviewed article.

Some minor reviews should bring the study to level suitable of publication in a standard medical journal.

Abstract

1. Line 31- Begin methods with From July…….

Response: This change has been made (see line 31)

2. Any particular reason for the age-range and why the general population was sampled?

Response: The study was targeting consenting adults and +18y are considered as such in Malawi. The general population was included in order to understand the feasibility of use among untrained users who would be the likely target for large scale use at scale.

3. Did the rest of the participants identify as male?

Response: Yes, 656 (54%) participants were male (see line 35-36)

4. Conclusion should focus on the study. 

Response: Changes have been made to reflect this. “One of the first studies to demonstrate high feasibility and acceptability of self-testing using SARS-CoV-2 Ag-RDTs among both general and health-care worker populations in low- and middle-income countries potentially supporting large scale-up. Further research is warranted to provide optimal delivery strategies of self-testing.” (see 55-58)

5. Keywords 

Response: Keywords have been added (see line 59)

Introduction:

6. Acknowledge that the testing gap between HIC and LMICs is slowly reducing 

Response: Reference added. However, the trend throughout most of the pandemic was consistent with the HICs testing more than LMICs and this has largely remained as such. We wish to highlight this discrepancy and while we appreciate the reviewer’s comment we feel that it is worth highlighting that LMICs have continued to face challenges with testing as well as care. (see line 67-68)

7. Remove or remodify phrase on line 60

Response: The phrase has now been removed as requested by the reviewer. (see line 70)

8. Use less invasive example (line 67)

Response: We are grateful for the suggestion by the reviewer and we have now modified the statement as follows: “Self-testing is not a new paradigm with pregnancy self-testing and HIV self-testing being successful examples”. (Lines 79-80)

Methods: 

9. Improve description of setting 

Response: Description of setting has been improved including the rationale behind the site choice. A map has also been added. (see line 122-130)

10. Qualitative data analysis: Include description of how cognitive data was analysed

Response: A paper describing the qualitative process in detail is in the pipelines. That is why this is not described in this paper

Results 

11. Were IFUs presented in local language? Did this have any impact?

Response: IFUs were translated into Chichewa. Participants either provided feedback on the content during cognitive interviews or used the translated IFUs during the observational studies

Discussion

12. This section needs to mention the crucial place for IFUs in the large-scale success of self-sampling and self-testing. There will be no demonstrators for the people who will be using the kits 

In a setting such as our own, where there are low literacy rates and low exposure to technology, the accurate implementation of self-testing at large would still require trained volunteers or personnel to support the participants. However, there is still a place for culturally relevant IFUs for those that can read. (see line 295-297) 

Reviewer #2: 

Title: COVID-19 self-testing using antigen rapid diagnostic tests: feasibility evaluation among health-care workers and general population in Malawi

Well-done to the authors for such a robust work on Covid-19 self-testing and self sampling

Abstract

1. In the background and throughout the document- The use of COVID-19 and SARS-CoV-2 interchangeably is confusing. The context in which they are used can be more defined. I will suggest using COVID-19 for self-testing and sampling instances and SARS-CoV-2 for the rapid diagnostic test. For example, COVID-19 self-testing and COVID-19 sampling and then SARS-CoV-2 Antigen-Rapid Diagnostic Tests (Ag-RDTs).

Response: Changes have been made in the document to reflect the concern above (see line 27 and line 148)

2. Line 31- Begin methods with From July…….

Response: This change has been made (see line 31)

3. Line 33- In the methods are the Standard Q and Panbio tests registered trademarks? If so indicate with the symbol®

Response: They are not registered trademarks yet.

4. Line 34- randomly or non-random sampling?

Response: Participants were non randomly sampled (see line 34 and 139-140)

5. Line 48-49-The conclusion can include summary points for usability and acceptability of the tests. Also, implications for practice and policies?

Response: Summary points on acceptability and implications for practice and policy added (see line 55-58)

6. Line 61- are the SARS-CoV-2 rapid antigen tests (Ag-RDTs) recommended? By who? Or do you mean they are specifically useful to complement….

Response: The WHO does recommend the use of RDTs to complement NAATs. 

(see https://apps.who.int/iris/bitstream/handle/10665/345948/WHO-2019-nCoV-Antigen-Detection-2021.1-eng.pdf?sequence=1&isAllowed=y)

7. Line 64- would be good to provide refs for these testing setbacks in resource limited setting

Response: References provided (see line 74)

8. Line 65- provide a definition of self-testing and self-sampling in the context in which it is being used in this paper

Response: Definition provided (see line 77-79)

9. The introduction could be further strengthened re- what is the acceptability and feasibility of COVID-19 self-testing reported in HICs? Have there been any reported use of COVID-19 self-tests in other low resource settings prior? What is the general argument with regards to feasibility and acceptability of COVID-19 self-tests especially in LMICs?

Response: Additional points/references on acceptability and feasibility of self-testing in HICs added (see line 95)

There were no reports of Covid-19 self-tests in other low resource settings prior to the work according to our knowledge. (see line 95-97) 

Work on HIV self-testing in Malawi did point to the fact that COVID-19 self-testing could be highly feasible. The important addition being a need for an in-person demonstration as part of the process. 

10. Methods: Line 86- were the five components or phases conducted one after the other or in parallel or was there any specific order?

Response: The five components were conducted one after the other and in the order in which they are listed. A statement has been added to clarify this. (see line 113-114)

11. Line 104- how were the participants systematically sampled from out-patients? In-person during attendance? Via registers?

Response: Participants were non randomly sampled. We have changed to this from systematically sampled as this was wrongly used (see line 34 and 139-140)

12. Line 126- Was participant consent obtained for audio recording?

Response: Yes. Participants provided consent for each of the components of the study. (see line 136)

13. Line 155- How was the qualitative data from interviews analysed and presented?

Response: We have a separate paper reporting the results of the cognitive interviews and in-depth interviews. That is why this was not described in this paper

14. Line 134- Was there any bias introduced from researcher interpretation of interview data? Or influence in the results? Any steps taken to address reflexivity in the qualitative aspects?

Response: We have a separate paper reporting the results of the cognitive interviews and in-depth interviews. That is why this was not described in this paper

Results

15. Line 182- Best to start with total number of participants included in the study and then present the breakdown of participants per component. Starting with a total of 120 participants and then going up to 664 recruited from 732 screened is confusing

Response: A statement providing the total number of participants recruited across the 5 components has been included. We hope this will provide more clarity. (see line 213)

16. Line 183 how many were recruited from each site and how many were selected for interviews?

Response: Table 1 and 2 provides the breakdown of recruitment per site. (see page 11 and 12)

17. Line 189- How many participants were in the cognitive interviews?

Response: 120 participants were recruited in the cognitive interviews. (See line 214)

18. Line 228- Nearly is how many out of how many total interviewed?

Response: We have now removed the word nearly to “all” after confirming that it was all who were interviewed (line 268).

Discussion

19. Line 237- one of the first studies to the best of your knowledge?

Response: This statement “This is one of first studies to the best of our knowledge” has been added (see line 280)

20. Line 242- is it Panbio or PanBio? Review and maintain consistency throughout the manuscript.

Response: It is Panbio. Changes have been made throughout the manuscript to reflect this. 

21. Line 258- I don’t really see a small number of positive results as a limitation but a strength how was this managed? Linkage to care? If shows the tests were administered correctly

Response: Thank you for pointing this out. Participants were linked to the Ministry of Health services for counseling, management and contact tracing as was policy at the time. 

22. Line 264- were any steps taken to reduce the impact on extreme assessors?

Response: More than seven assessors were involved in the rating of the participants. We believe that this could have helped in minimizing the bias (see 178-179)

23. Line 266- Are there others strengths of this work? Originality?

Response: Thank you for pointing this out. Indeed, originality was a strength of this study and has been included. (see line 280)

24. Line 274- What are the key recommendations? Any implications for practice policies and future research?

Response: A statement on implications on policy and further research has been added (see line 331-335)

25. Line 276- missing “of” in 2 places

Response: This has been rectified (see line 326)

26. Line 277- LMICs if defined earlier no need to spell out in full

Response: This has been noted and corrected (see line 327)

---

## [Decision Letter · Decision Letter 1]

17 Jul 2023

COVID-19 self-testing using antigen rapid diagnostic tests: feasibility evaluation among health-care workers and general population in Malawi

PONE-D-22-25536R1

Dear Dr. Mukoka, 

We’re pleased to inform you that your manuscript has been judged scientifically suitable for publication and will be formally accepted for publication once it meets all outstanding technical requirements.

Kind regards,

Gheyath K. Nasrallah

Academic Editor

PLOS ONE

Additional Editor Comments (optional):

Reviewers' comments:

Reviewer's Responses to Questions

**Comments to the Author**

1. If the authors have adequately addressed your comments raised in a previous round of review and you feel that this manuscript is now acceptable for publication, you may indicate that here to bypass the “Comments to the Author” section, enter your conflict of interest statement in the “Confidential to Editor” section, and submit your "Accept" recommendation.

Reviewer #1: All comments have been addressed

2. Is the manuscript technically sound, and do the data support the conclusions?

Reviewer #1: Yes

3. Has the statistical analysis been performed appropriately and rigorously? 

Reviewer #1: Yes

4. Have the authors made all data underlying the findings in their manuscript fully available?

Reviewer #1: Yes

5. Is the manuscript presented in an intelligible fashion and written in standard English?

Reviewer #1: Yes

6. Review Comments to the Author

Reviewer #1: Dear Author,

Thank you for this addition to the body of knowledge as we navigate the COVID-19 experience in the world of science.

The use of self-sampling and testing method is already at top gear in most of the high economy countries, however, no addition is counted obsolete in infectious diseases.

I am happy that the necessary review has been done and I think the study is worthy of publication at the moment.

Congratulations.

7. PLOS authors have the option to publish the peer review history of their article (what does this mean?). If published, this will include your full peer review and any attached files.

Reviewer #1: **Yes: **Adeloye Amoo Adeniji (MBBS; MMed; FCFP;FACRRM)

---

## [Editor Report · Acceptance letter]

21 Jul 2023

PONE-D-22-25536R1 

COVID-19 self-testing using antigen rapid diagnostic tests: feasibility evaluation among health-care workers and general population in Malawi 

Dear Dr. Mukoka:

I'm pleased to inform you that your manuscript has been deemed suitable for publication in PLOS ONE. Congratulations! Your manuscript is now with our production department. 

Kind regards, 

on behalf of

Dr. Gheyath K. Nasrallah 

Academic Editor

PLOS ONE